# Utility of Rapid Molecular Assays for Detecting Multidrug-Resistant *Mycobacterium tuberculosis* in Extrapulmonary Samples

**DOI:** 10.3390/diagnostics15091113

**Published:** 2025-04-28

**Authors:** Katarzyna Kania, Katarzyna Wójcik, Kamil Drożdż, Karolina Klesiewicz

**Affiliations:** 1Department of Pharmaceutical Microbiology, Faculty of Pharmacy, Jagiellonian University Medical College, ul. Medyczna 9, 30-688 Krakow, Poland; ka.kania@uj.edu.pl; 2Laboratory of Microbiology, The St. John Paul II Specialist Hospital, ul. Pradnicka 80, 31-202 Krakow, Poland; 3Malopolska Central Laboratory of Tuberculosis Diagnostics, The St. John Paul II Specialist Hospital, ul. Ulanow 29, 31-455 Krakow, Poland; k.wojcik@szpitaljp2.krakow.pl; 4Department of Molecular Medical Microbiology, Faculty of Medicine, Jagiellonian University Medical College, ul. Czysta 18, 31-121 Krakow, Poland; kamil.drozdz@uj.edu.pl

**Keywords:** drug-resistant tuberculosis, molecular diagnostics, pulmonary, extrapulmonary TB

## Abstract

**Background**: Extrapulmonary tuberculosis (TB) presents significant diagnostic challenges, particularly in the context of multidrug-resistant (MDR) strains. This study assessed the utility of the WHO-recommended rapid molecular assays, originally validated for pulmonary TB, in diagnosing extrapulmonary TB and detecting the MDR *Mycobacterium tuberculosis* complex (MTBC). **Materials and Methods**: A total of 6274 clinical samples, including 4891 pulmonary and 1383 extrapulmonary samples, were analyzed between 2019 and 2022 using the BD MAX™ MDR-TB assay (BD MAX), the Xpert^®^ MTB/RIF assay (Xpert MTB/RIF), the Xpert^®^ MTB/XDR assay (Xpert MTB/XDR), FluoroType MTB, and phenotypic drug susceptibility testing (DST). **Results**: MTBC was detected in 426 samples using BD MAX (376 pulmonary and 50 extrapulmonary), of which 277 were culture-confirmed. Phenotypic testing confirmed 299 positive cultures on Löwenstein–Jensen (LJ) medium and 347 in BD BACTEC™ MGIT™ (BACTEC MGIT) mycobacterial growth indicator tube (BBL) liquid culture. BD MAX showed high sensitivity and specificity for extrapulmonary TB detection (93.1% and 98.4%, respectively). Resistance to isoniazid or rifampicin was identified in 11% of MTBC-positive cases, whereas 3.69% were confirmed as MDR-TB. The molecular assays effectively detected resistance-associated mutations *(katG*, *inhA*, and *rpoB*), with high concordance to phenotypic tests (DST) (κ = 0.69–0.89). **Conclusions**: This study demonstrates that molecular assays, although validated for pulmonary TB, are also reliable for extrapulmonary TB detection and drug resistance profiling. Their rapid turnaround and robust accuracy support broader implementation in routine diagnostics, especially for challenging extrapulmonary specimens where early detection is critical for targeted therapy.

## 1. Introduction

Tuberculosis (TB), caused by the *Mycobacterium tuberculosis* complex (MTBC), remains one of the deadliest infectious diseases worldwide. Each year, over 10 million people develop TB, with approximately 1.3 million deaths [1]. In 2022, TB remained the second leading cause of mortality from a single infectious agent—surpassed only by the COVID-19 pandemic—and caused nearly twice as many deaths as HIV/AIDS [2]. 

According to the WHO recommendations, the first-line therapy for drug-susceptible TB, both pulmonary TB and extrapulmonary TB, consists of isoniazid, rifampicin, ethambutol, and pyrazinamide [3]. The standard regimen lasts six months, with all four drugs administered during the first two months, followed by isoniazid and rifampicin for the remaining four months. Despite the availability of effective treatment guidelines, drug-resistant TB remains a significant public health challenge. The prevalence of multidrug-resistant tuberculosis (MDR-TB) continues to pose a major threat to global TB control efforts [4]. 

Rifampicin is one of the most potent anti-TB drugs, targeting both rapidly and slowly growing *M. tuberculosis* bacilli, making it a cornerstone of drug-susceptible TB treatment regimens. The rapid adoption of molecular diagnostic technologies has significantly improved TB detection and the identification of rifampicin resistance. However, a major concern is the high prevalence of TB strains resistant to isoniazid but susceptible to rifampicin, which often leads to underdiagnosed cases [5]. 

The diagnosis of drug-resistant TB still relies on antimicrobial susceptibility testing. Due to the slow growth of *M. tuberculosis*, phenotypic drug susceptibility testing can take up to eight weeks, thereby delaying appropriate treatment. During this period, patients often receive ineffective therapy, increasing the risk of further drug resistance. Therefore, it is crucial to promptly identify drug resistance to avoid treatment failure and improve patient outcomes [3]. 

Over the past two decades, molecular techniques for detecting TB drug resistance have advanced significantly. Nucleic acid amplification tests (NAAT) allow for rapid diagnosis within hours of sample collection. Although most commercially available molecular assays have been primarily validated for pulmonary TB, data on their use in extrapulmonary samples remain limited. Although culture remains the gold standard for TB diagnosis [6], new-generation molecular tests, such as automated real-time PCR assays, demonstrate high sensitivity. These assays not only detect *M. tuberculosis* but also identify resistance markers for rifampicin and isoniazid, with sensitivity comparable to that of culture-based methods [7,8,9].

Isoniazid resistance is primarily associated with mutations in the *katG* gene, which impair drug activation, or mutations in regulatory regions, such as *inhA*. Rifampicin resistance is predominantly caused by mutations in the *rpoB* gene, altering the drug’s target site on RNA polymerase and reducing its effectiveness [8].

Molecular assays for detecting resistance to key anti-TB drugs in clinical laboratories and healthcare settings have significantly enhanced treatment effectiveness. Compared with phenotypic drug susceptibility testing, which requires several weeks, molecular methods can provide critical resistance profiles within hours, allowing for more timely and targeted therapy [3]. Despite the high sensitivity and effectiveness of molecular assays in TB diagnosis, not all tests have been validated for the detection of extrapulmonary TB. This study conducted a comprehensive molecular assessment of TB infections in southern Poland.

This study aimed to highlight the advantages of molecular techniques in providing rapid and reliable diagnostics, particularly in extrapulmonary samples, where culture-based methods are limited by a low bacterial load. Their ability to detect *M. tuberculosis* and resistance markers within hours supports timely and targeted treatment.

Our study also aimed to compare commercially available rapid molecular tests with phenotypic methods for *M. tuberculosis* in extrapulmonary samples within routine diagnostics. In addition, we evaluated the drug resistance mechanisms of MTBC strains and compared the performance of four molecular assays: the BD MAX™ MDR-TB assay (BD MAX), the Xpert^®^ MTB/RIF Ultra assay (Xpert MTB/RIF), the Xpert^®^ MTB/XDR assay (Xpert MTB/XDR), and FluoroType^®^ MTB (FluoroType MTB). Furthermore, we aimed to highlight the advantages of molecular techniques in delivering rapid and reliable results to support a more effective and tailored treatment for patients with TB. 

## 2. Materials and Methods

### 2.1. Study Samples 

This study is a retrospective analysis based on samples processed as part of routine diagnostics at the Malopolska Central Laboratory of Tuberculosis Diagnostics, St. John Paul II Specialist Hospital, Krakow, Poland. No additional assays were performed beyond the standard laboratory procedures. This study included clinical samples collected between September 2019 and June 2022, which were processed strictly according to established diagnostic protocols. 

The N-acetyl-L-cysteine sodium hydroxide decontamination method was used to effectively liquefy and decontaminate the clinical samples. The obtained sediments were resuspended in 1.5 mL of phosphate buffer and cultured on the LJ solid medium (incubated at 35 ± 1 °C for up to 8 weeks) and in BD BACTEC™ MGIT™ mycobacterial growth indicator tubes (BBL) (BD Diagnostics, Franklin Lakes, NJ, USA) containing liquid medium, incubated in the BD BACTEC™ MGIT™ 960 (BACTEC MGIT) (BD Diagnostics, Franklin Lakes, NJ, USA) mycobacterial detection system for up to 42 days.

Simultaneously, smears were prepared from the sediments and stained with auramine for fluorescence microscopy, while the Ziehl–Neelsen method was used as a confirmatory technique for positive smears. 

The complete methodological workflow is presented in Figure 1 in Appendix A.

### 2.2. Phenotypic Drug Susceptibility Test 

#### 2.2.1. Susceptibility Testing for Anti-TB Agents Using Liquid Media 

The MTBC-confirmed isolates were tested for susceptibility to the following drugs utilizing the MGIT system: streptomycin, isoniazid, rifampicin, ethambutol, and pyrazinamide. The phenotypic DST was conducted in accordance with the WHO recommendation [1]. The BACTEC MGIT system used modified Middlebrook 7H9 broth (7 mL). For this study, the lyophilized BACTEC MGIT 960 SIRE drugs (streptomycin: 332.0 µg, isoniazid: 33.2 µg, and ethambutol: 1660.0 µg) were dissolved in 4 mL of distilled water, rifampicin: 332.0 µg was dissolved in 8 mL of distilled water, and pyrazinamide in 2.5 mL of distilled water. For drug susceptibility testing, 7 mL of the BBL medium was supplemented with 0.8 mL of the SIRE reagent and inoculated with 0.5 mL of MTBC culture, diluted in a ratio of 1:5 in physiological saline, following the manufacturer’s recommendations. Each tube received 0.1 mL of the respective drug solution (streptomycin, isoniazid, rifampicin, and ethambutol). For pyrazinamide testing, the medium was supplemented with 0.8 mL of this drug and 0.1 mL of the pyrazinamide solution. The final drug concentrations were as follows: isoniazid 0.1 µg/mL, rifampicin 0.5 µg/mL, ethambutol 5.0 µg/mL, streptomycin 1.0 µg/mL, and pyrazinamide 100 µg/mL (in PZA medium, pH 5.9) [3,10,11].

All samples were loaded into the BACTEC MGIT system and incubated for up to 13 days for streptomycin, isoniazid, rifampicin, and ethambutol, and up to 21 days for pyrazinamide. Growth in the drug-containing tubes was compared with that in a drug-free control tube once the latter reached a predefined threshold, expressed in growth units.

#### 2.2.2. Susceptibility Testing for Anti-TB Agents Using Solid Media

To confirm drug resistance detected with the BACTEC MGIT system, the conventional agar proportion method on a drug-containing medium was applied. The LJ medium was prepared in accordance with WHO recommendations [3,10,11] and contained critical concentrations of anti-tuberculosis drugs. The medium was then inoculated with a diluted culture suspension. The proportion method was employed to develop protocols for the DST of first-line anti-tuberculosis agents. To apply the 1% criterion, the control inoculum was diluted 100-fold relative to the inoculum in the drug-containing tube. The anti-TB agents recommended for DST using the LJ medium and established critical concentrations are as follows: isoniazid: 0.2 µg/mL, rifampicin: 40.0 µg/mL, ethambutol: 2.0 µg/mL, amikacin: 30.0 µg/mL, ethionamide: 40 µg/mL, and streptomycin: 4.0 µg/mL. Ofloxacin at 0.5 µg/mL and rifabutin at 0.5 µg/mL were tested on Middlebrook 7H10 agar (BD Diagnostics, Franklin Lakes, NJ, USA) [3,10].

### 2.3. Rapid Molecular Detection of M. tuberculosis and Isoniazid and Rifampicin Resistance

#### 2.3.1. BD MAX MDR-TB Assay 

The assays using the BD MAX system were conducted in accordance with the manufacturer’s recommendations [12]. Following the decontamination of the clinical samples, the sediment was prepared. The samples were brought to room temperature and 1 mL of each was used for the analysis. BD MAX STR (sample treatment reagent) was added to each sample in a 2:1 ratio, followed by agitation and incubation at room temperature for 30 min. A 2.5 mL volume of the resulting suspension was then transferred to a BD MAX MDR-TB tube for further processing with test strips and master mix tubes. Results were categorized as MTB detected, MTB low positive, negative, or not detected, with simultaneous mutation analysis for rifampicin (targeting the *rpoB* gene, positions 507–533) and isoniazid (targeting the *katG* and *inhA* genes) [Table 1]. 

#### 2.3.2. GenoType MTBDR Plus 

Samples that tested positive for mutations associated with resistance to rifampicin and/or isoniazid had their drug susceptibility profiles confirmed using the GenoType MTBDRplus assay (Hain Lifescience GmbH, Bruker, Nehren, Germany). The analysis was performed directly on a culture of MTBC isolates grown on LJ solid medium. Rifampicin resistance was identified by detecting key mutations in the *rpoB* gene (codons 505–533). Isoniazid resistance was determined by examining the *katG* gene and the promoter region of the *inhA* gene [Table 1]. 

Bacterial colonies grown on LJ media served as the initial material for DNA extraction according to the manufacturer’s recommendations [13]. The A and B mixes (AM-A and AM-B) of GenoType MTBDRplus (Hain, Lifescience GmbH, Bruker, Nehren, Germany) were used for amplification. The master mix was prepared in a DNA-free environment after DNA extraction with GenoLyse (Hain, Lifescience GmbH, Bruker, Nehren, Germany). Each 50 μL sample contained 10 μL AM-A, 35 μL AM-B, and 5 μL of the tested DNA solution. The amplification schedule for clinical specimens was as follows: 15 min at 95 °C for 1 cycle, 30 s at 95 °C, 2 min at 65 °C for 20 cycles, 25 s at 95 °C, 40 s at 50 °C, 40 s at 70 °C for 30 cycles, and finally 8 min at 70 °C for 1 cycle. Hybridization was manually performed using the TwinCubator with pre-warmed reagents (37–45 °C). For each strain, 20 µL of the denaturation solution was mixed with 20 µL of the amplified product and incubated at room temperature for 5 min. The strips were immersed in 1 mL hybridization buffer and incubated at 45 °C for 30 min with shaking. Having removed the buffer with a vacuum pump, the strips were washed with 1 mL of the stringent wash solution and incubated at 45 °C for 15 min. For detection, the strips were rinsed with 1 mL of rinse solution for 1 min, then incubated with 1 mL of diluted conjugate solution at room temperature for 30 min with shaking. After 2 rinses with the rinse solution and 1 with distilled water, 1 mL of the diluted substrate solution was added and incubated in the dark until the bands appeared (3–20 min). The reaction was stopped by rinsing with distilled water. The strips were dried between the absorbent paper layers and evaluated with the provided template. The result assessment and analysis were conducted following the manufacturer’s guidelines [13]. 

#### 2.3.3. GeneXpert MTB/RIF

The Xpert MTB/RIF Ultra (Cepheid, Sunnyvale, CA, USA) assay was performed on the samples that had tested positive for mutations associated with rifampicin resistance. NAAT was used to detect the presence of MTBC and the rifampicin-resistance-associated mutations of the *rpoB* gene [Table 2]. The method employed the GeneXpert Instrument System with a disposable cartridge. All procedures were performed according to the manufacturer’s instructions [14]. Briefly, 1 mL of the positive BBL culture was mixed with 2 mL of SR reagent, vortexed for 10 s, and incubated for 10 min. After re-vortexing, the sample was left to stand for 5 min and then loaded into the Xpert MTB/RIF cartridge for the analysis. 

#### 2.3.4. GeneXpert MTB/XDR 

Mutations related to isoniazid and second-line drug resistance were evaluated using the automated GeneXpert MTB/XDR system (Cepheid, Sunnyvale, CA, USA) from positive cultures in accordance with the manufacturer’s instructions [15]. The samples were prepared analogously to the GeneXpert MTB/RIF method. The assay detected MTBC genetic material and identified the following mutations associated with isoniazid resistance in the *katG* and *fabG1* genes and the *inhA* promoter, mutations in the *gyrA* and *gyrB* regions conferring fluoroquinolone resistance, and mutations in the *rrs* gene and *eis* promoter region linked to resistance to second-line injectable drugs [Table 1]. 

### 2.4. Quality Control 

The *M. tuberculosis* ATCC 27294 (MTB H37Rv) reference strain served as the quality control (QC) for both the phenotypic DST method and the NAAT assays. This QC strain is sensitive to all the first- and second-line drugs evaluated in this study. 

### 2.5. Statistics and Data Analysis 

The statistical analysis was performed using PQStats 1.8.6.102. The parameters for diagnostic tests detecting MTBC were calculated along with the 95% confidence interval (CI). The agreement of rifampicin and isoniazid gene detection results was determined using the Kappa–Cohen concordance test. *p*-values < 0.05 were considered significant. 

### 2.6. Limitations 

This study has several limitations. First, while the microbiological analysis of isolates, including species identification and drug resistance profiling, was comprehensive, no clinical or radiological data were collected or analyzed. Although patients may have undergone radiographic imaging or other clinical assessments as part of routine care, this information was not included in our analysis.

### 2.7. Ethics Statement 

Due to the retrospective nature of this study, the requirement for obtaining informed consent was waived by the Bioethics Committee of the Regional Medical Chamber in Krakow. All experimental protocols were approved by the Bioethics Committee of the Regional Medical Chamber in Krakow (L.dz.OIL/KBL/1/2025) and the Committee for Ethics of Scientific Research of the Jagiellonian University Collegium Medicum in Krakow (118.0043.1.19.2025). 

## 3. Results

### 3.1. Detection of M. tuberculosis in Pulmonary and Extrapulmonary Samples

In total, 6274 samples were tested, including 4891 pulmonary and 1383 extrapulmonary specimens [Table 2]. These samples were collected as part of routine diagnostics over a four-year period: 545 in 2019, 1453 in 2020, 1792 in 2021, and 2484 in 2022. The 1383 extrapulmonary specimens included pleural fluid (n = 826), lymph node biopsies (n = 188), tumor punctures (n = 117), intraoperative specimens (n = 80), urine (n = 64), gastric lavage (n = 56), other tissue samples (n = 30), cerebrospinal fluid (n = 17), and vertebral bone specimens (n = 5). All samples were submitted for differential diagnosis to confirm or exclude tuberculosis and were tested using both molecular and, where applicable, phenotypic methods.

#### 3.1.1. Phenotypic Assays

A total of 6274 clinical samples were tested using phenotypic diagnostic methods, including acid-fast bacilli (AFB) smear microscopy, solid culture on LJ medium, and BBL liquid culture. Of all the samples, 258 were positive by AFB staining, 299 showed growth on the LJ medium, and 347 tested positive in the BBL culture [Table 3].

When stratified by specimen type, 233 pulmonary samples were AFB-positive, compared with 25 extrapulmonary ones. Growth on the LJ medium was observed in 269 pulmonary and 30 extrapulmonary samples. The BBL method demonstrated the highest detection yield, with 311 positive pulmonary and 36 extrapulmonary samples.

#### 3.1.2. Molecular Assay for the Detection of *M. tuberculosis*

Between 2019 and 2022, 6274 samples were analyzed, including 4891 pulmonary and 1383 extrapulmonary ones. Positive results for MTBC were confirmed in 6.78% (426/6274) of the examined specimens based on the BBL. In 2019, MTBC genetic material was detected in 41 samples, in 82 in 2020, in 154 in 2021, and in 149 samples in 2022 [Figure 1]. 

During the study period, MTBC genetic material was detected in 425 of 6274 samples (6.77%) using the BD MAX assay. Culture on the LJ medium confirmed growth in 277 of these cases, while 140 samples were BD MAX-positive but culture-negative. Conversely, 23 samples showed growth on LJ despite negative BD MAX results.

In pulmonary samples (*n* = 4891), MTBC was detected in 376 cases (7.69%), including 249 confirmed by LJ culture and 118 without culture confirmation. Growth without genetic confirmation was observed in 21 cases.

In extrapulmonary samples (*n* = 1383), MTBC was detected in 49 cases (3.54%), including 27 confirmed by LJ culture and 22 that were BD MAX-positive only. Growth without genetic confirmation was observed in two cases. Additionally, twenty-five extrapulmonary samples were positive only in the BBL, and seven samples were excluded due to contamination [Table 4].

Moreover, a statistical analysis was performed to demonstrate the high sensitivity and specificity of the BD MAX molecular test and compare the results of pulmonary and extrapulmonary samples (>90%, 95% CI) [Table 5]. For patients with confirmed MTBC growth on the LJ solid medium (the gold standard), the BD MAX test exhibited a sensitivity of 92.3% (95% CI: 88.6–95.0%) and specificity of 97.7% (95% CI: 97.2–98.0%) among participants with negative LJ cultures. The sensitivity for pulmonary and extrapulmonary samples was 92.2% (95% CI: 88.3–95.1%) and 93.1% (95% CI: 77.2–99.2%), respectively, while their specificity equaled 97.4% (95% CI: 96.9–97.9%) and 98.4% (95% CI: 97.6–99.0%), respectively. BD MAX’s sensitivity in the AFB smear-positive samples was 69.9% (95% CI: 64.4–75.0%) of the total, with 70.6% (95% CI: 64.8–76.0%) for pulmonary samples and 63.3% (95% CI: 43.9–80.1%) for extrapulmonary ones. 

In our study, we compared the sensitivity and specificity of the BD MAX system, the BBL method, and AFB smear microscopy across all the tested samples (pulmonary and extrapulmonary). Overall, BD MAX demonstrated high values across the sample types, confirming its value in TB diagnostics. It showed a sensitivity of 92.3% (95% CI: 88.6–95.0%) and a specificity of 97.7% (95% CI: 97.2–98.0%). However, BBL exhibited slightly higher sensitivity (99.7%, 95% CI: 98.2–100%) and specificity (99.3%, 95% CI: 99.1–99.5%), reflecting its status as a well-established phenotypic diagnostic approach. The AFB smear had the lowest sensitivity (69.9%, 95% CI: 64.4–75.0%), though its specificity remained high at 99.5% (95% CI: 99.3–99.6%). These findings highlight the limitations of AFB in detecting TB in cases with low bacterial loads, particularly when compared to molecular or culture-based methods. 

For pulmonary samples, the BD MAX system performed similarly to the overall cohort, with a sensitivity of 92.2% (95% CI: 88.3–95.1%) and a specificity of 97.4% (95% CI: 96.9–97.9%). BBL again demonstrated slightly superior sensitivity (99.6%) and specificity (99.3%). In contrast, AFB had significantly lower sensitivity (70.6%, 95% CI: 64.8–76.0%), confirming its limited reliability as a stand-alone diagnostic tool, particularly in smear-negative but culture-positive cases. 

In the extrapulmonary samples, all tested methods displayed slightly lower accuracy, likely due to their paucibacillary nature. However, BD MAX maintained good sensitivity (93.1%, 95% CI: 77.2–99.2%) and specificity (98.4%, 95% CI: 97.6–99.0%). BBL demonstrated excellent sensitivity (100%, 95% CI: 88.4–100%) and specificity (99.6%, 95% CI: 99.0–99.8%). As expected, AFB had the lowest sensitivity in the extrapulmonary samples (63.3%, 95% CI: 43.9–80.1%), further emphasizing its limitations in detecting TB in specimens with a low bacterial burden. However, its specificity remained high at 99.6%. 

The analysis of the positive predictive value (PPV) and negative predictive value (NPV) confirmed the reliability of BD MAX. The test had an overall PPV of 66%, which was slightly lower than that of BBL (88%), likely due to its molecular ability to detect nonviable organisms, which may result in false positives in previously treated or latent infections. The NPV of BD MAX was nearly perfect at 1.00 (95% CI: 0.99–1.00), indicating that a negative BD MAX result is highly reliable for ruling out TB. Similarly, BBL had an NPV of 1.00, while AFB had a slightly lower NPV (0.99), meaning that although a negative smear is generally reliable, it is not as definitive as molecular or culture-based methods. 

The positive likelihood ratio (LR+) for BD MAX was 39.38, indicating that a positive result significantly increased the probability of TB. This was lower than BBL (148.83) but comparable to AFB smear microscopy (134.75). The negative likelihood ratio (LR−) of BD MAX was 0.08, strongly suggesting that a negative result significantly reduces the probability of TB. However, it was slightly higher than BBL’s (0.003), reflecting the latter’s superior sensitivity. In the extrapulmonary samples, BD MAX performed particularly well, with an LR+ of 57.39 and an LR− of 0.07, demonstrating its robustness in diagnostically challenging cases. 

The overall BD MAX accuracy was 97.4% (95% CI: 97.0–97.8%), which was slightly lower than that of BBL (99.3%) but significantly higher than that of the AFB smear (98.1%). In the pulmonary samples, BD MAX maintained 97.2% accuracy, again slightly lower than BBL but still demonstrating strong performance. In extrapulmonary cases, BD MAX showed an excellent accuracy of 98.3%, confirming its usefulness in detecting tuberculosis even in difficult-to-diagnose cases. 

### 3.2. Susceptibility Testing

#### 3.2.1. Phenotypic Drug Susceptibility Testing

Phenotypic DST was performed in accordance with the WHO recommendations using two methods: the automated BACTEC MGIT 960 system and the conventional agar proportion method on LJ solid medium. The automated method was used to detect resistance to streptomycin, isoniazid, rifampicin, and ethambutol, while the agar proportion method was applied to assess susceptibility to rifampicin, isoniazid, amikacin, ofloxacin, rifabutin, ethambutol, ethionamide, and streptomycin.

The results obtained using the BACTEC MGIT system showed that among the 347 tested strains, 308 strains (88.76%) were susceptible to the tested drugs, while 39 strains (11.23%) were resistant to at least one of the tested anti-mycobacterial agents [Table 6].

Strains resistant to rifampicin and/or isoniazid were confirmed by the conventional agar proportion method on the solid medium used as a gold standard [Figure 2].

Resistance to at least isoniazid or rifampicin, the key drugs in TB treatment, was exhibited by 39 isolates. Rifampicin resistance was observed in 18 isolates and isoniazid resistance in 32 isolates.

Among the analyzed strains, we confirmed 17 isolates as MDR-TB. Seven strains were identified as pre-extensively drug-resistant (Pre-XDR) TB, with resistance to isoniazid, rifampicin, and ofloxacin. Four strains were classified as extensively drug-resistant (XDR) TB, showing resistance to these three drugs as well as additional second-line drugs.

#### 3.2.2. Susceptibility Testing Using Molecular Methods

Drug susceptibility testing was performed using the BD MAX assay to detect the most common mutations associated with resistance to rifampicin or isoniazid. A total of 39 samples out of 307 were analyzed. These samples tested positive, indicating MTBC genetic material and *rpoB* and/or *katG*/*inhA* promoter mutations, as detected by the BD MAX assay. The samples that yielded low positive results (119 samples) were excluded due to the assay’s insufficient sensitivity for mutation detection in such cases. Among the tested samples, resistance to rifampicin was confirmed in 21 cases and resistance to isoniazid in 34 cases [Table 7]. 

In addition, we compared the most commonly used methods for detecting clinically relevant drug resistance. Samples positive for MTBC were further analyzed using two widely applied molecular assays: GenoType MTBDRplus (Hain, Lifescience GmbH, Bruker, Nehren, Germany) and the GeneXpert MTB/XDR system (Cepheid, Sunnyvale, CA, USA), as well as two phenotypic methods based on the automated BACTEC™ MGIT™ 960 system (BD Diagnostics, Franklin Lakes, NJ, USA) and agar proportion on solid media susceptibility testing [Table 8]. 

The analysis of rifampicin resistance across the different assays revealed varying detection rates. BD MAX identified the highest number of rifampicin-resistant isolates, detecting 21 cases. In comparison, the GenoType MTBDRplus and BACTEC MGIT assays each detected 17 isolates, representing the lowest number of rifampicin-resistant isolates among the assays. The Xpert MTB/XDR assay identified 18 rifampicin-resistant isolates, while the agar proportion method detected 19. These differences in the number of resistant isolates could be attributed to the varied sensitivity and specificity of the molecular and phenotypic assays used for resistance detection.

The detection of isoniazid-resistant isolates was more consistent across the assays. BD MAX detected 34 of these, while GenoType MTBDRplus, GeneXpert MTB/XDR, and BACTEC MGIT each identified 33. The agar proportion method detected 32 isoniazid-resistant isolates, i.e., slightly fewer than the molecular methods. The minimal variability in isoniazid-resistant detection across the assays suggests a higher agreement between the molecular and phenotypic methods in identifying isoniazid-resistant isolates compared with rifampicin resistance. We also analyzed the co-occurrence of resistance to both isoniazid and rifampicin. The drug susceptibility profile for isoniazid and rifampicin obtained via BD MAX is shown in Figure 3.

The concordance of the different methods employed to determine drug susceptibility to rifampicin and isoniazid was evaluated in comparison to the BD MAX system. Statistical analysis was conducted to assess the concordance between methods and effectively detect resistance to isoniazid and rifampicin. Table 9 presents substantial or great agreement between the methods. All of them showed substantial to almost perfect agreement with BD MAX for rifampicin and isoniazid susceptibility, with kappa values ranging from 0.69 to 0.89. The agar proportion method demonstrated the highest agreement for rifampicin susceptibility (κ = 0.80), while both MTBDRplus and GeneXpert MTB/XDR showed almost perfect agreement for isoniazid (κ = 0.89). All the methods had statistically significant concordance with BD MAX for both rifampicin and isoniazid, as indicated by the *p*-values (<0.001). 

## 4. Discussion

Drug-resistant tuberculosis remains a major global health threat, affecting half a million people annually. The global prevalence of drug-resistant TB is estimated to be approximately 11.6% [16]. Managing drug-resistant TB is challenging because it requires longer and more complex therapy regimens, often with poorer outcomes. Despite this, only a third of patients have access to high-quality care, underscoring the urgent need for improved diagnostics and treatment strategies [2,16]. 

According to current WHO definitions [17], strains of *M. tuberculosis* complex with an MDR profile (MDR/RR-TB) demonstrate resistance to at least isoniazid and rifampicin. Pre-XDR-TB is defined as MDR/RR-TB strains that are also resistant to fluoroquinolone, which is crucial in tuberculosis treatment, while XDR-TB is classified as TB caused by MTBC strains meeting MDR/RR-TB criteria and demonstrating resistance to fluoroquinolones and at least one additional Group A drug (levofloxacin, moxifloxacin, bedaquiline, or linezolid). Swift identification of drug resistance is critical for initiating appropriate treatment. 

Our study confirms the high diagnostic performance of WHO-endorsed molecular assays for extrapulmonary TB, showing that BD MAX achieved a sensitivity of 93.1% and a specificity of 98.4% in detecting MTBC from extrapulmonary specimens. These values are comparable or slightly superior to those reported by Behera et al., who evaluated Xpert MTB/RIF in extrapulmonary samples and observed sensitivities ranging from 71% to 90%, depending on the sample type and bacillary load [1]. Furthermore, our detection rate for rifampicin resistance (6.5%) is consistent with global averages and comparable to findings by Armstrong et al., who assessed the BD MAX MDR-TB assay and confirmed its reliable performance for both MTBC detection and resistance profiling in non-sputum samples [2]. Importantly, our analysis included comparative results from BD MAX, as well as from Xpert MTB/RIF, Xpert MTB/XDR, and FluoroType MTB assays, supported by phenotypic testing, allowing a comprehensive evaluation. The high concordance between the molecular and phenotypic results in our study (κ = 0.69–0.89) underscores the reliability of molecular tools in routine diagnostics, particularly for extrapulmonary TB, where conventional culture is often limited by low bacterial loads. These findings support the broader use of molecular assays beyond pulmonary TB and align with prior studies highlighting their impact on the early detection and management of drug-resistant tuberculosis [18,19].

Our study analyzed the drug resistance profiles of MTBC strains, focusing on resistance to first-line and second-line drugs according to the WHO criteria [17]. Based on their resistance patterns, the strains were classified as MDR-TB, pre-XDR-TB, or XDR-TB. In our dataset, 17 strains were MDR-TB and resistant to both isoniazid and rifampicin, highlighting the significant challenge of drug-resistant TB in clinical management. In addition, seven strains exhibited resistance to ofloxacin, meeting the criteria for pre-XDR-TB. The emergence of pre-XDR-TB strains emphasizes the necessity of alternative treatment regimens, as fluoroquinolones are essential in managing drug-resistant TB. Furthermore, four strains were classified as XDR-TB.

Our analysis, conducted between 2019 and 2022, preceded the current WHO definitions. Consequently, we did not perform tests regarding susceptibility to newer drugs, such as linezolid, bedaquiline, moxifloxacin, and levofloxacin. According to the WHO recommendations, the first-line treatment for drug-sensitive TB (both pulmonary and extrapulmonary) consists of a six-month regimen of four antibiotics: isoniazid, rifampicin, ethambutol, and pyrazinamide [2,3]. While this regimen remains effective for most cases, our analysis found that 91.84% of the strains from southern Poland were susceptible to all first-line drugs. Therefore, resistance to isoniazid (8.8%) and rifampicin (6.8%) highlights the need for tailored treatment strategies and close monitoring. Resistance to these key drugs is often a marker for MDR-TB, necessitating second-line treatments [20]. 

MDR-TB, present in nearly 4% of the cases in our study, is consistent with global trends and underscores the need for timely intervention [21,22]. Although phenotypic drug susceptibility testing remains the gold standard for detecting drug resistance [17], it has several limitations. It is time-consuming, requires specialized expertise and infrastructure, and is less effective in extrapulmonary TB because of the typically low bacterial loads in these samples [23,24]. Despite the challenges, phenotypic DST in our study demonstrated high sensitivity (93.1%) and specificity (98.4%) for extrapulmonary samples, confirming its diagnostic accuracy when bacterial loads were sufficient. These findings align with previous studies by Ciesielczuk et al. [25]. 

Given the limitations of phenotypic DST, particularly in low-bacterial-load cases such as extrapulmonary TB, alternative or complementary methods must be explored. Molecular diagnostics, such as molecular drug susceptibility testing, offers faster results and can differentiate between low-level and high-level isoniazid resistance by identifying specific mutations. This enables a more precise treatment in certain cases, such as for high-dose isoniazid [21]. Molecular assays, including BD MAX, GeneXpert Ultra, and FluoroType MTBDR, significantly improved TB diagnostics with more rapid and reliable results than traditional culture-based methods [26]. 

Local monitoring of drug resistance is essential for adapting treatment strategies to current resistance patterns. Our analysis revealed that between 2019 and 2022, nearly 8% of diagnosed MTBC strains in southern Poland were resistant to at least isoniazid or rifampicin, with MDR-TB present in 3.69% of cases. NAAT, endorsed by the WHO, detects well-characterized resistance mutations in *katG* and the *inhA* promoter, which are frequently associated with isoniazid resistance [11]. Globally, isoniazid resistance occurs in approximately 8% of rifampicin-susceptible TB cases, further underscoring the need for rapid and accurate detection methods [27]. 

Molecular diagnostics have transformed TB management by enabling faster drug resistance detection. In particular, BD MAX was thoroughly investigated and demonstrated high sensitivity and specificity in detecting isoniazid and rifampicin resistance [18,28]. This makes it an essential tool for diagnosing drug-resistant TB, facilitating timely decisions to mitigate the spread of resistance. Notably, BD MAX detects mutations in the *inhA*, *katG*, and *rpoB* genes, distinguishing it from other molecular tests that primarily focus on rifampicin resistance [29]. Accurate and timely identification of isoniazid resistance is crucial for tailoring treatment regimens. The WHO estimates that 8% of TB patients worldwide are resistant to both of them. In regions where isoniazid mono-resistance exceeds 10–20%, treatment outcomes are compromised when standard regimens are used [1,30]. As such, the rapid detection of isoniazid resistance, facilitated by BD MAX, can prevent the progression to additional resistance and improve patient outcomes [22,26,30,31,32]. 

As the prevalence of MDR-TB, pre-XDR-TB, and XDR-TB continues to grow, enhanced TB management strategies are urgently need, such as the development of novel drugs and improved diagnostic tools [26,28]. In our study, molecular diagnostics, such as BD MAX, demonstrated effective performance in diagnosing both pulmonary and extrapulmonary TB, with a sensitivity and specificity above 90%. However, continued validation of these tools is necessary, particularly for extrapulmonary cases where diagnostic challenges are more pronounced [33]. 

## 5. Conclusions

This study confirms that WHO-endorsed molecular assays, although originally validated for pulmonary samples, perform reliably in extrapulmonary tuberculosis diagnostics. They demonstrated high sensitivity and specificity, with strong concordance to phenotypic methods, even in low-bacillary-load specimens where culture is often limited. Our findings highlight the value of incorporating molecular tools into routine diagnostics for extrapulmonary TB, supporting the earlier detection of drug resistance and more effective clinical management. As drug-resistant TB remains a pressing global challenge, expanding access to rapid and accurate diagnostic methods will be key to improving treatment outcomes and curbing transmission.

## Figures and Tables

**Figure 1 diagnostics-15-01113-f001:**
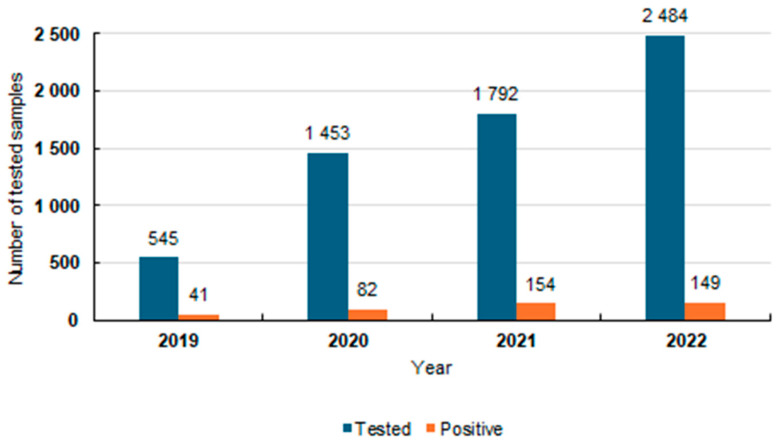
Analysis of tested samples between 2019 and 2022 with positive results for MTBC via the BD MAX assay.

**Figure 2 diagnostics-15-01113-f002:**
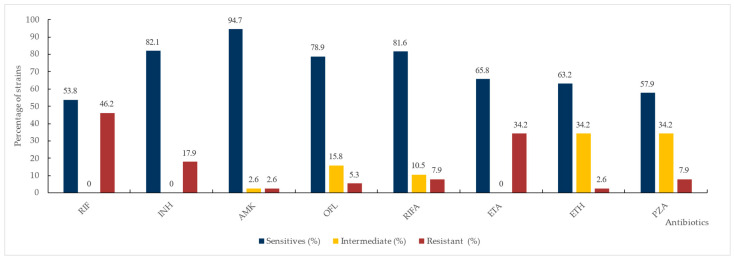
Susceptibility profile of MTBC isolates against the most common anti-mycobacterial drugs via the agar proportion method. RIF—rifampicin, INH—isoniazid, AMK—amikacin, OFL—ofloxacin, RIFA—rifabutin, ETA—ethambutol, ETH—ethionamide.

**Figure 3 diagnostics-15-01113-f003:**
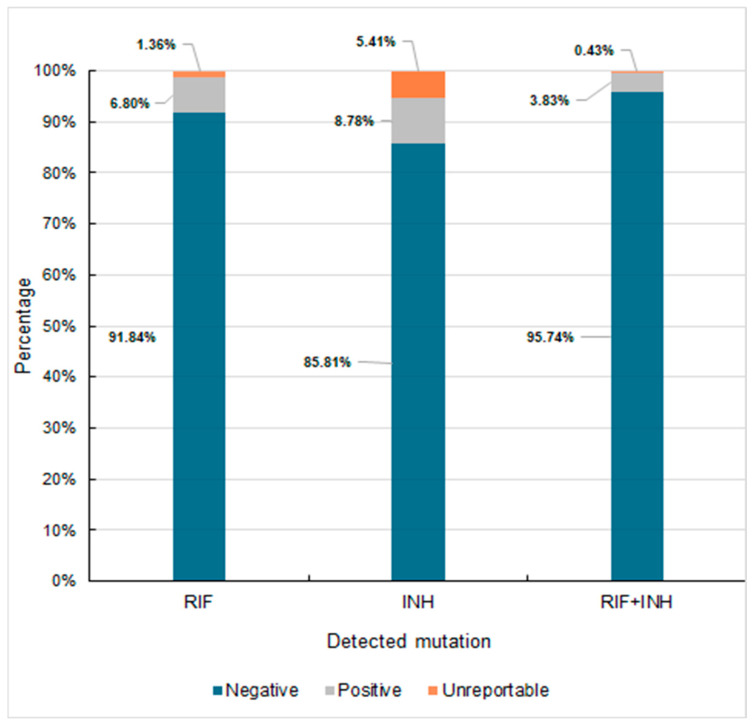
Percentage distribution of mutations for rifampicin and isoniazid among the tested strains of MTB. RIF—rifampicin, INH—isoniazid.

**Table 1 diagnostics-15-01113-t001:** Summary of regions, nucleotide positions, and mutations detected by different MDR-TB molecular diagnostic tests [9,12,13,14,15].

Test	Gene	Codon Range	Targeted Mutation Region	Identified Amino Acid Alterations
Hain MTBDRplus	*rpoB*	505–509	Rifampicin Resistance Determining Region (RRDR)	F505L, T508A, S509T
		510–513		Q510H, L511P
		510–517		Q513L, Q513P, del514-516
		513–319		D516V, D516Y, Del515,
		516–522		Del518, N518,
		518–525		S522L, S522Q
		526–529		H526Y, H526D, H526R, H526P, H526Q, H526N, H526L, H526S, H526C,
		530–533		S531L, S531Q, S531W, L533P,
	*katG*	944–945	Isoniazid Resistance	
	*inhA*	−15, −8	Isoniazid Resistance Promoter	C15T
GeneXpert^®^ MTB/RIF	*rpoB*	426–452	Rifampicin Resistance Determining Region (RRDR)	
BD MAX MDR-TB	*rpoB*	507–533	Rifampicin Resistance Determining Region (RRDR)	
	*katG*	944–945	Isoniazid Resistance	
	*inhA*	*inhA* promoter region	Isoniazid Promoter	
Xpert^®^ MTB/XDR	*katG*	315	Isoniazid Resistance	codon 315, S315T
	*fabG1*	199–210	fabG1 promoter region	G609A
	*oxyR–ahpC*	5 to −50, −47 to −92	Intergenic region between oxyR and ahpC	
	*inhA*	1 to −32	Isoniazid and Ethionamide Resistance Promoter	C15T
	*gyrA*, *gyrB*	87–95, 531–544	Fluoroquinolone Resistance	D94G, A90V
	*rrs*	1396–1417	Resistance to Second-Line Drugs Amikacin, Kanamycin, and Capreomycin	A1401G
	*eis*	6 to −42	eis promoter	G-10A, C-12T

**Table 2 diagnostics-15-01113-t002:** Clinical materials analyzed using the BD MAX system from 2019 to 2022, categorized based on collection site.

Sample Obtained for Testing		No. of Tested Samples
**Pulmonary samples**		**4891**
	Bronchoalveolar Lavage (BAL)	413
	Sputum	1018
	Bronchial wash	3460
**Extrapulmonary samples**		**1383**
	Pleural fluid	826
	Lymph node biopsy	188
	Tumor puncture	117
	Intraoperative specimen	80
	Urine	64
	Gastric lavage	56
	Tissue sample (other)	30
	Cerebrospinal fluid	17
	Vertebral bone specimen	5
**Total**		**6274**

**Table 3 diagnostics-15-01113-t003:** Comparison of phenotypic assay results for pulmonary and extrapulmonary samples, including detection rates and statistical significance (*n* = 6274).

Samples	AFB-Positive *n* (%)	LJ-Positive *n* (%)	BBL-Positive *n* (%)
Total (*n* = 6274)	258 (4.11%)	299 (4.77%)	347 (5.53%)
Pulmonary (*n* = 4891)	233 (3.55%)	269 (4.28%)	311 (4.95%)
Extrapulmonary (*n* = 1383)	25 (0.39%)	30 (0.48%)	36 (0.57%)

**Table 4 diagnostics-15-01113-t004:** Summary of MTBC detection results using the genetic probe and culture methods in pulmonary and extrapulmonary samples.

Samples	Positive Results for the Genetic Probe	Positive Results for the Genetic Probe with Confirmed Growth on LJ Media	Positive Results for the Genetic Probe with No Confirmation on LJ Growth	Growth on LJ Solid Media with Negative Genetic Probe Assay
Total (*n* = 6274)	425	277	140	23
Pulmonary (*n* = 4891)	376	249	118	21
Extrapulmonary (*n* = 1383)	49	27	22	2

*n*—number of samples.

**Table 5 diagnostics-15-01113-t005:** Comparison of the sensitivity and specificity of the methods used to confirm the presence of MTBC in clinical samples, using genetic tests (BD MAX), BBL culture, and AFB direct microscopy vs. culture on LJ solid medium as the gold standard.

Tested Parameter	Total			Pulmonary Samples			Extrapulmonary Samples		
	BD MAX	BBL	AFB	BD MAX	BBL	AFB	BD MAX	BBL	AFB
TP	275	298	209	248	268	190	27	30	19
FP	140	40	31	118	34	25	22	6	6
FN	23	1	90	21	1	79	2	0	11
TN	5835	5933	5945	4501	4582	4594	1334	1351	1351
Sensitivity (95% CI)	92.3% (88.6–95.0%)	99.7% (98.2–100%)	69.9% (64.4–75%)	92.2% (88.3–95.1%)	99.6% (97.9–100%)	70.6% (64.8–76.0%)	93.1% (77.2–99.2%)	100% (88.4–100%)	63.3% (43.9–80.1%)
Specificity (95% CI)	97.7% (97.2–98.0%)	99.3% (99.1–99.5%)	99.5% (99.3–99.6%)	97.4% (96.9–97.9%)	99.3% (99.0–99.5%)	99.5% (99.2–99.6%)	98.4% (97.6–99.0%)	99.6% (99.0–99.8%)	99.6% (99.0–99.8%)
PPV (95% CI)	0.66 (0.61–0.71)	0.88 (0.84–0.91)	0.87 (0.82–0.91)	0.68 (0.63–0.73)	0.89 (0.85–0.92)	0.88 (0.83–0.92)	0.55 (0.40–0.69)	0.83 (0.67–0.94)	0.76 (0.55–0.91)
NPV (95% CI)	1.00 (0.99–1.00)	1.00 (1.00–1.00)	0.99 (0.98–0.99)	1.00 (0.99–1.00)	1.00 (1.00–1.00)	0.98 (0.98–0.99)	1.00 (0.99–1.00)	1.00 (1.00–1.00)	0.99 (0.99–1)
LR (+) (95% CI)	39.38 (33.33–46.54)	148.83 (109.27–202.70)	134.75 (94.11–192.93)	36.09 (30.10–43.27)	135.26 (96.76–189.08)	130.50 (87.61–194.38)	57.39 (37.47–87.88)	226.17 (101.79–502.53)	143.24 (61.62–332.97)
LR (-) (95% CI)	0.08 (0.05–0.12)	0.003 (0.001–0.026)	0.30 (0.25–0.36)	0.08 (0.05–0.12)	0.004 (0.001–0.0027)	0.30 (0.25–0.36)	0.07 (0.02–0.27)	-	0.37 (0.23–0.59)
ACC (95% CI)	97.4% (97.0–97.8%)	99.3% (99.1–99.5%)	98.1% (97.7–98.4%)	97.2% (96.7–97.6%)	99.3% (99–99.5%)	97.9% (97.4–98.3%)	98.3% (97.4–98.9%)	99.6% (99.1–99.8%)	98.8% (98.0–99.3%)

TP—true positive, FP—false positive, FN—false negative, TN—true negative, CI—confidence interval, PPV—positive predictive value, NPV—negative predictive value, LR+ = positive likelihood ratio; LR− = negative likelihood ratio, ACC—Accuracy.

**Table 6 diagnostics-15-01113-t006:** Results of susceptibility testing of 347 M. tuberculosis strains positive on BBL culture obtained using the automated BACTEC MGIT 960 system.

Anti-Tuberculosis Drug	Number of Resistant Isolates (%)
Streptomycin	27 (7.78%)
Isoniazid	30 (8.65%)
Rifampicin	19 (5.48%)
Ethambutol	12 (3.46%)
Pyrazinamide	14 (4.03%)

**Table 7 diagnostics-15-01113-t007:** Results of M. tuberculosis complex resistance profiles obtained from BD MAX MDR-TB assays between 2019–2022.

Results of the BD MAX MDR-TB Assay	All Tested Samples			Percentage Value (%)		
Total	T (*n* = 6274)	P (*n* = 4891)	EX (*n* = 1383)	T [%]	P	EX
MTBC ND	5848	4515	1333	93.21	92.31	96.38
MTBC (TOTAL)	426	376	50	6.79	7.68	3.62
MTBC L-P in TOTAL	119	103	16	1.90	2.10	1.16
RIF R-ND	270	240	30	4.30	4.9	2.17
RIF R-D	21	20	1	0.33	0.41	0.072
RIF R-UNR	135	116	19	2.15	2.37	1.37
INH R-ND	254	225	29	4.0	4.6	2.10
INH R-D	27	24	3	0.43	0.49	0.22
*katG* Mut-D	22	-	-	0.35	-	-
*inhApr* Mut-D	14	-	-	0.22	-	-
INH R-UNR	145	127	18	2.31	2.6	1.3

*n*—number of samples, MTBC—Mycobacterium tuberculosis complex, ND—not detected, L-P—low positive, R-ND—resistance not detected, R-D—resistance detected, R-UNR—resistance unreportable, RIF—rifampicin, INH—isoniazid, S—sensitive, R—resistance, *kat*G Mut-D—mutations present in the *kat*G assay target, *inhApr*—mutations present in the *inh*A promoter assay. T—total, P—pulmonary, EX—extrapulmonary.

**Table 8 diagnostics-15-01113-t008:** Summary of the resistance results for rifampicin and isoniazid performed using different assays.

Assay Performed	No. of Isolates RIF-R	No. of Isolates INH-R
BD MAX	21	34
GenoType MTBDRplus	17	33
GeneXpert MTB/XDR	18	33
BACTEC MGIT	17	34
AP	18	32

BD MAX MDR-TB Assays (Becton Dickinson), GenoType MTBDRplus (Hain, Lifescience GmbH, Bruker, Nehren, Germany), and the GeneXpert^®^ MTB/XDR system (Cepheid, Sunnyvale, CA, USA), as well as two phenotypic methods based on the BACTEC MGIT 960 automatic system (BD Diagnostics, Franklin Lakes, NJ, USA) and agar proportion (AP) on solid media susceptibility testing.

**Table 9 diagnostics-15-01113-t009:** Concordance of the four methods used to determine drug susceptibility to isoniazid and rifampicin, evaluated in comparison to the BD MAX method.

Test Applied	Results	RIF		κ	*p*	INH		κ	*p*
		Positive	Negative			Positive	Negative		
GenoType MTBDRplus	Positive	15	1	0.69	<0.001	33	0	0.89	<0.001
	Negative	5	18			1	5		
GeneXpert MTB/XDR	Positive	16	1	0.74	<0.001	33	0	0.89	<0.001
	Negative	4	18			1	5		
BACTEC MGIT	Positive	16	1	0.74	<0.001	33	1	0.77	<0.001
	Negative	4	18			1	4		
AP	Positive	17	1	0.80	<0.001	32	0	0.80	<0.001
	Negative	3	18			2	5		

κ ranged from 0.61 to 0.8 for substantial agreement and was greater than 0.8 for great agreement. BD MAX MDR—TB Assays (Becton Dickinson), GenoType MTBDRplus (Hain, Lifescience GmbH, Bruker, Nehren, Germany), and the GeneXpert MTB/XDR^®^ system (Cepheid, Sunnyvale, CA, USA), as well as two phenotypic methods based on the BACTEC MGIT 960 automatic system (BD Diagnostics, Franklin Lakes, NJ, USA) and agar proportion (AP) on solid media susceptibility testing.

## Data Availability

All data analyzed or generated during this study are published in the article.

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
