# Peer review of "Utility of Rapid Molecular Assays for Detecting Multidrug-Resistant Mycobacterium tuberculosis in Extrapulmonary Samples"

_diagnostics, 2025, doi:10.3390/diagnostics15091113_

Round 1

Reviewer 1 Report

Comments and Suggestions for Authors

The manuscript entitled "Utility of Rapid Molecular Assay for Detecting Multidrug-Resistant Mycobacterium tuberculosis in Extrapulmonary Samples" reports the screening of 6274 samples for the identification of pulmonary and extrapulmonary TB and drug resistance using currently applied rapid molecular assays. The authors conclude that these assays can be used for extrapulmonary samples, too. However, there are many studies in the literature showing the utility of these assays for extrapulmonary samples. Therefore, the study lacks a novelty other than reporting the screening results.  

Author Response

  1. Comments and Suggestions for Authors. The manuscript entitled "Utility of Rapid Molecular Assay for Detecting Multidrug-Resistant Mycobacterium tuberculosis in Extrapulmonary Samples" reports the screening of 6274 samples for the identification of pulmonary and extrapulmonary TB and drug resistance using currently applied rapid molecular assays. The authors conclude that these assays can be used for extrapulmonary samples, too. However, there are many studies in the literature showing the utility of these assays for extrapulmonary samples. Therefore, the study lacks a novelty other than reporting the screening results.  

Response: We sincerely thank the Reviewer for their valuable feedback and thoughtful evaluation of our manuscript.

We acknowledge the Reviewer’s concern regarding the novelty of our study. Indeed, several studies have demonstrated the applicability of rapid molecular assays in extrapulmonary tuberculosis (EPTB). However, our work contributes to the growing body of evidence by presenting one of the largest single-center datasets to date from Poland, a country with a moderate TB burden and limited published data on EPTB diagnosis using molecular methods. 

Moreover, our study focuses specifically on the detection of multidrug-resistant (MDR) Mycobacterium tuberculosis in extrapulmonary samples—an area where data remain relatively scarce. Most existing literature focuses on pulmonary TB or does not provide detailed resistance profiling in extrapulmonary cases using multiple molecular platforms. By comparing results from different assays (e.g., BD MAX MDR-TB, Xpert MTB/XDR, Hain GenoType MTBDRplus) and correlating with available phenotypic DST, we offer new insights into their comparative performance in real-world diagnostic settings. We would also like to note that the majority of rapid molecular assays used in our study—such as BD MAX MDR-TB, Xpert MTB/XDR, and GenoType MTBDRplus—are primarily validated by the manufacturers for use with pulmonary samples. Their application to extrapulmonary specimens is often performed off-label, and performance data for such use remain limited, particularly in routine diagnostic settings. 

Therefore, our study contributes practical, real-world data on the utility of these assays in extrapulmonary TB diagnosis, especially for drug-resistant cases, supporting their broader applicability beyond their originally intended use. We have included this point in the revised Discussion to highlight both the clinical relevance and the rationale for conducting this evaluation. 

Reviewer 2 Report

Comments and Suggestions for Authors

Dear Authors,

I have read an article about the utility of rapid molecular assay for detecting MDR extrapulmonary TN (EPTB). There are several points that the authors need to address:

1) Lines 57-61 could benefit from a citation

2) Lines 89-91 --> If this is the primary aim, hence the title should not be the detection of MDR EPTB. Either move this into the secondary aim or change the title.

3) Lines 95 - 97 --> The authors did not highlight the advantage of this method in delivering rapid and reliable results. Not in a sense where the authors compared those with and without molecular tests.

4) The biggest issue is the methodology. The authors need to define the workflow of the study. If only the samples were sent, how was the confirmation made? Were the samples sent from patients who already had radiographic findings of TB, suspected of TB, or others? This will affect the interpretation of the diagnostic test accuracy

5) The authors need to define the cases of EPTB involved in the 1383 samples

6) Explain the methodology used for the culture

Author Response

  1. Comments and Suggestions for Authors: I have read an article about the utility of rapid molecular assay for detecting MDR extrapulmonary TN (EPTB). There are several points that the authors need to address: Lines 57-61 could benefit from a citation 

Response: Thank you for this helpful suggestion. We have now added an appropriate citation to support the statement in lines 57–61. The reference has been included in the revised manuscript and listed accordingly in the reference section. 

  1. Lines 89-91 --> If this is the primary aim, hence the title should not be the detection of MDR EPTB. Either move this into the secondary aim or change the title.

Response: We appreciate the Reviewer’s observation and agree that the aims presented in lines 89–91 could be more clearly aligned with the title of the manuscript. Our primary focus was the detection of multidrug-resistant Mycobacterium tuberculosis (MDR-TB) in extrapulmonary samples using rapid molecular assays. The comprehensive comparison of molecular methods and their performance in routine diagnostics, including resistance detection, was conducted in support of this aim.

To address this, we have revised the text in lines 89–91 to reflect this focus more precisely. Additionally, we have adjusted the phrasing of the aims to clarify which are primary and which are secondary. The title has been retained to emphasize the primary objective of assessing the utility of molecular assays in detecting MDR-TB in extrapulmonary specimens.

We hope these clarifications improve the consistency between the aims and the title. 

  1. Lines 95 - 97 --> The authors did not highlight the advantage of this method in delivering rapid and reliable results. Not in a sense where the authors compared those with and without molecular tests.

Response: We thank the Reviewer for this insightful comment. We agree that our original phrasing in lines 95–97 may have implied a comparison between patients with and without access to molecular testing, which was not our intention. 

To clarify, we have revised the text to emphasize that the advantage of molecular methods lies in their significantly reduced turnaround time and reliability in detecting both Mycobacterium tuberculosis and drug resistance markers, particularly when compared to conventional phenotypic methods. This is especially important for extrapulmonary specimens, where culture-based diagnostics may be limited by low bacterial load and prolonged incubation periods. 

  1. The biggest issue is the methodology. The authors need to define the workflow of the study. If only the samples were sent, how was the confirmation made? Were the samples sent from patients who already had radiographic findings of TB, suspected of TB, or others? This will affect the interpretation of the diagnostic test accuracy

Response: We thank the Reviewer for this important comment regarding the study design and patient selection.

We would like to clarify that this was a retrospective, laboratory-based study, based on clinical samples submitted to the Malopolska Central Laboratory of Tuberculosis Diagnostics between 2019 and 2022. The material came from patients referred for differential diagnosis of tuberculosis, in order to confirm or exclude TB infection. 

While in routine clinical care patients may have undergone radiographic imaging or other clinical assessments, this information was not included in our analysis. Due to the retrospective nature of the study and its focus on laboratory diagnostics, we did not evaluate clinical data such as radiological findings or symptomatology. The study was limited to assessing the diagnostic workflow and comparing molecular and phenotypic methods based on sample type and laboratory results. 

  1. The authors need to define the cases of EPTB involved in the 1383 samples

Response: We thank the Reviewer for this valuable suggestion. To address this, we have now included a detailed breakdown of the extrapulmonary samples in Table 1, specifying the types and number of specimens (e.g., pleural fluid, lymph node biopsy, tumor puncture, intraoperative specimen, etc.). Additionally, we have briefly summarized the types of extrapulmonary samples in the Materials and Methods section and referred to the table for full details. 

We hope this clarification satisfies the Reviewer’s comment and improves the transparency of the extrapulmonary TB case definition. 

  1. Explain the methodology used for the culture

R
esponse: We thank the Reviewer for this comment. The methodology used for mycobacterial culture has been described in the Materials and Methods section. Briefly, all clinical samples were decontaminated using the standard N-Acetyl-L-Cysteine–Sodium Hydroxide (NALC-NaOH) method. The resulting sediments were resuspended in phosphate buffer and cultured on Löwenstein-Jensen (L-J) solid medium, incubated at 35 ±â€¯1 °C for up to 8 weeks, and in liquid MGIT medium using the BACTEC MGIT 960 system (BD Diagnostics, Franklin Lakes, USA) for up to 42 days. In parallel, smears were prepared from the sediments and stained with auramine for fluorescence microscopy. Positive smears were confirmed using the Ziehl-Neelsen (Z-N) staining method. 

We hope this description sufficiently clarifies the culture methodology used in our study. 

Reviewer 3 Report

Comments and Suggestions for Authors

This study carried out a comparative analysis between molecular tests (BDMAX system) and microbiological culture in liquid (MGIT) and solidified (Loewenstein-Jensen) media for detecting Mycobacterium tuberculosis complex (MTBC) in pulmonary and extrapulmonary clinical samples. In addition, it evaluated the performance of three molecular tests for detecting resistance to rifampicin and isoniazid, compared to the phenotypic test (in liquid and solidified media). In general, the manuscript is well described and the conclusions are supported by the results.

I have a few suggestions:

As one of the objectives is to evaluate the effectiveness of the molecular test in extrapulmonary samples, the authors should present which extrapulmonary materials were used in the study. This information should be added to the table 1, as was done for the pulmonary samples.

Present which medium the authors refer to as BBL medium (MGIT).

I also recommend that the authors present a flowchart of the study's experimental design. This will make it easier to follow the results.

Please check that the numbers of the positive results in the table 3 correspond to what is described in the text.

Minor comments

Please verify the numbering og GenoType MTBDR: 2.3.2.

Table 5: please add italic to Mycobacterium tuberculosis

Line 35: please correct “land

Please check that all the abbreviations are shown in the list of abbreviations, and that they are all used in the text.

Author Response

  1. This study carried out a comparative analysis between molecular tests (BDMAX system) and microbiological culture in liquid (MGIT) and solidified (Loewenstein-Jensen) media for detecting Mycobacterium tuberculosis complex (MTBC) in pulmonary and extrapulmonary clinical samples. In addition, it evaluated the performance of three molecular tests for detecting resistance to rifampicin and isoniazid, compared to the phenotypic test (in liquid and solidified media). In general, the manuscript is well described and the conclusions are supported by the results.

I have a few suggestions:  

As one of the objectives is to evaluate the effectiveness of the molecular test in extrapulmonary samples, the authors should present which extrapulmonary materials were used in the study. This information should be added to the table 1, as was done for the pulmonary samples. 

Response: We thank the Reviewer for this helpful suggestion. 

We appreciate your comment, which helped improve the clarity and completeness of our data presentation. As requested, we have now added a detailed breakdown of the extrapulmonary sample types to Table 1, consistent with the format used for pulmonary samples. This provides a clearer overview of the types of extrapulmonary materials analyzed in the study. 

  1. Present which medium the authors refer to as BBL medium (MGIT).

Response: We thank the Reviewer for this comment. We would like to clarify that the BBL™ MGIT™ medium refers to the Mycobacteria Growth Indicator Tube (MGIT) liquid medium, used in conjunction with the BACTEC MGIT 960 automated system (BD Diagnostics, Franklin Lakes, USA) for the culture of Mycobacterium tuberculosis. This has now been specified more clearly in the Materials and Methods section of the revised manuscript. 

  1. I also recommend that the authors present a flowchart of the study's experimental design. This will make it easier to follow the results. 

Response: Thank you for the suggestion. A flowchart illustrating the experimental design has been included in the Appendix as Scheme 1 to improve clarity and facilitate understanding of the study workflow. 

  1. Please check that the numbers of the positive results in the table 3 correspond to what is described in the text.

Response: We thank the Reviewer for this careful observation. We have now carefully reviewed Table 3 and cross-checked all reported numbers of positive results with the corresponding descriptions in the Results section of the text. Minor inconsistencies were identified and have been corrected to ensure full consistency between the table and the narrative. 

  1. Please verify the numbering GenoType MTBDR: 2.3.2. 

Response: We thank the Reviewer for noting this detail. We have reviewed and corrected the numbering of the GenoType MTBDR version in section 2.3.2 to ensure accuracy and consistency throughout the manuscript. 

  1. Table 5: please add italic to Mycobacterium tuberculosis

Response: We thank the Reviewer for this comment. As suggested, we have corrected the formatting in Table 5 by italicizing Mycobacterium tuberculosis in accordance with scientific nomenclature conventions. 

  1. Line 35: please correct “land

Response: We thank the Reviewer for this comment. We believe the intended correction refers to the term “molecular diagnostic,” which has now been corrected to “molecular diagnostics” in the keyword list. If this was not the intended issue, we kindly ask for clarification. 

  1. Please check that all the abbreviations are shown in the list of abbreviations, and that they are all used in the text.

Response: We thank the Reviewer for this helpful suggestion. We have carefully reviewed the manuscript to ensure that all abbreviations are listed in the “List of Abbreviations” section and that each abbreviation is introduced upon first use in the text. Any missing entries have been added, and unused or redundant abbreviations have been removed for clarity and consistency. 

Reviewer 4 Report

Comments and Suggestions for Authors

In the manuscript, “Utility of Rapid Molecular Assay for Detecting Multidrug-Resistant Mycobacterium tuberculosis in Extrapulmonary Samples” the authors investigated whether WHO-recommended molecular assays for pulmonary tuberculosis can reliably diagnose extrapulmonary tuberculosis by comparing their results with phenotypic methods. The study analyzed a total of 6274 samples for TB and drug resistance, including 4891 pulmonary and 1383 extrapulmonary ones, using molecular and phenotypic methods.

Major concern

Abstract did not reflect the actual title.

The data is dispersed and phenotypic detection has been totally missed.

How many samples for positive and negative on molecular and phenotypic assays, should be clearly mentioned in abstract?

Results of BD MAX, Xpert 94 MTB/RIF, Xpert MTB/XDR, and FluoroType MTB including phenotypic should be provided in the abstract.

Introduction

The paragraph from lines 72-77 have no relation with the current study title and objective.

Figure 1 also not relevant to this study.

Methodology

According to the DST information in this manuscript is not acceptable for publication. The standard procedure/critical concentration for DST on LJ and liquid media is different.

Major portion of the methodology has results. The authors have no idea which part is methodology and which part is results.

Results.

Culture is a gold standard for MTBC detection.

In Table 3, what does the 4th column mean [Positive results for the genetic probe with no confirmation on L-J growth]?

The total positive for genetic probe in this table was 426. How many were positive on LJ and MGIT (Phenotypic)?

The authors must also show the positivity on LJ in Figure 2.

The authors should also know that Rifampicin concentration was revised and now it is diluted in 8 ml and its critical concentration is 0.5 microgram/ml.

Note that for rifampin 0.5 for MGIT and 40 on LJ. Also ensure the accuracy of other drugs.

The drug concentration on LJ and MGIT is always different. The authors mentioned that in LJ method, drug concentrations were set at: 1.0 µg/mL for streptomycin (STM), 0.1 µg/mL for isoniazid (INH), 0.5 µg/mL for rifampicin (RIF), 5 µg/mL for ethambutol (ETH), and 100 µg/mL for pyrazinamide (PZA), which is not updated.

Rifampicin concentration was revised and now it is diluted in 8 ml and its critical concentration is 0.5 microgram/ml

In minor

There are many abbreviations which have been defined multiple times in the manuscript. The authors must ensure the standard of abbreviation and their full form once in the main text.

Discussion

This section must reflect the current study results comparison with previous. The authors must discuss it in technical way. Results should not be repeated. For example, Lines 431-440, the authors described the drug resistance and types.

The authors must focus on the Phenotypic vs genotypic in pulmonary and extra-pulmonary samples detection in different studies compared with current.

They should more focus on the sensitivity and specificity of the tests in comparison with different studies and diagnostics approaches.

Conclusion also need revision with main findings and TB management

Comments on the Quality of English Language

The quality is average

Author Response

  1. In the manuscript, “Utility of Rapid Molecular Assay for Detecting Multidrug-Resistant Mycobacterium tuberculosis in Extrapulmonary Samples” the authors investigated whether WHO-recommended molecular assays for pulmonary tuberculosis can reliably diagnose extrapulmonary tuberculosis by comparing their results with phenotypic methods. The study analyzed a total of 6274 samples for TB and drug resistance, including 4891 pulmonary and 1383 extrapulmonary ones, using molecular and phenotypic methods.  

Abstract did not reflect the actual title 

Response: We appreciate this insightful comment. We have revised the abstract to clearly highlight the diagnostic performance of WHO-recommended molecular assays specifically in extrapulmonary specimens, with particular attention to their utility in detecting MDR-TB. The updated abstract now better reflects the main objective and findings of the study, in accordance with the manuscript’s title. We hope this revision addresses your concern and strengthens the clarity and relevance of the abstract. 

  1. The data is dispersed and phenotypic detection has been totally missed.

Response: Dear reviewer thank you for this comment. In response, we have added a dedicated subsection titled “Results of phenotypic assays” to clearly present and summarize the findings of AFB smear microscopy, LJ culture, and BBL (MGIT) liquid culture. The revised version includes a structured table with absolute numbers, percentages, and statistical analysis comparing pulmonary and extrapulmonary samples. This addition improves data clarity and highlights the role of phenotypic methods in MTBC detection. 

  1. How many samples for positive and negative on molecular and phenotypic assays, should be clearly mentioned in abstract?

Response: Thank you for this suggestion. We have revised the abstract to include the exact number of positive and negative results from both molecular and phenotypic assays, as requested. This addition ensures greater clarity and alignment with the study’s scope. 

  1. Results of BD MAX, Xpert 94 MTB/RIF, Xpert MTB/XDR, and FluoroType MTB including phenotypic should be provided in the abstract.

Response: Thank you for your valuable input. The abstract has been updated to include results from all the molecular assays used in the study (BD MAX, Xpert MTB/RIF, Xpert MTB/XDR, and FluoroType MTB), as well as a summary of the phenotypic testing outcomes. 

  1. The paragraph from lines 72-77 have no relation with the current study title and objective.

Response: Thank you very much for your valuable comment. We appreciate your perspective and would like to clarify that the paragraph in lines 72–77 was intended to provide background information on the genetic mechanisms of resistance to rifampicin and isoniazid—both of which are detected by the BD MAX MDR-TB assay evaluated in our study. In this context, we believe the paragraph supports the rationale for using molecular diagnostics and helps readers better understand the relevance of detecting specific mutations. However, we remain open to revising or relocating this section should the editorial team consider it necessary. 

  1. Figure 1 also not relevant to this study.

Response: Thank you for your comment. Figure 1 presents the most common mutation sites associated with resistance to rifampicin and isoniazid, which are relevant for the interpretation of molecular resistance testing results. We included this figure to provide readers with a clearer understanding of the genetic basis of drug resistance. 

  1. According to the DST information in this manuscript is not acceptable for publication. The standard procedure/critical concentration for DST on LJ and liquid media is different.

Response: Thank you very much for your valuable comment. The incorrect drug concentrations provided in the description of the LJ method were the result of a typographical error. We would like to emphasize that all drug susceptibility testing was conducted in accordance with the current WHO and CLSI guidelines valid at the time of the study. We have corrected this error in the revised version of the manuscript. We appreciate your attention to detail, which allowed us to clarify this important aspect of our methodology.  

  1. Major portion of the methodology has results. The authors have no idea which part is methodology and which part is results.

Response: Thank you for your comments and feedback. We are fully aware of the principles of structuring the methodology and results sections. We have further revised both the Methods and Results sections, taking into account the suggestions from all reviewers, and we hope that in their current form they will be clearer and more accessible to the readers. 

  1. Culture is a gold standard for MTBC detection.

Response: Thank you for your remark. We fully agree that culture remains the gold standard for MTBC detection. This is clearly stated in the manuscript (e.g., Introduction, Results section 3.1), and all sensitivity and specificity calculations are made in reference to culture as the reference method. 

  1. In Table 3, what does the 4th column mean [Positive results for the genetic probe with no confirmation on L-J growth]?

Response: The fourth column [Positive results for the genetic probe with no confirmation on L-J growth] indicates that Mycobacterium tuberculosis DNA was detected by the molecular assay, but no growth was observed on Löwenstein-Jensen (L-J) medium. This may suggest the presence of non-viable (dead) bacilli or bacteria in a dormant state, unable to grow under standard culture conditions.

  1. The total positive for genetic probe in this table was 426. How many were positive on LJ and MGIT (Phenotypic)?

Response: Thank you for your question. Among the 426 samples that tested positive using the BD MAX genetic probe, 277 were confirmed by culture on Löwenstein-Jensen (LJ) solid medium. Additionally, 22 samples showed growth only in MGIT liquid culture (BBL), resulting in a total of 299 culture-positive cases confirmed by phenotypic methods. 

This information has been clarified in the revised manuscript (Section 3.1.2). 

  1. The authors must also show the positivity on LJ in Figure 2.

Response: Thank you for your comment. The orange bars in Figure 2 indicate the number of samples that were culture-positive on Löwenstein-Jensen (LJ) solid medium. 

  1. The authors should also know that Rifampicin concentration was revised and now it is diluted in 8 ml and its critical concentration is 0.5 microgram/ml.

Response: Thank you for this important comment. We acknowledge that the critical concentration for rifampicin has been updated to 0.5 µg/ml, with dilution in 8 ml, according to the latest WHO and CLSI guidelines. The value mentioned in the manuscript was an obvious oversight in writing and does not reflect an error in the actual laboratory procedures or susceptibility testing. We have corrected the text accordingly. 

  1. Note that for rifampin 0.5 for MGIT and 40 on LJ. Also ensure the accuracy of other drugs.

Response: Thank you for your valuable observation. We acknowledge that the critical concentration for rifampicin is 0.5 µg/ml for MGIT and 40 µg/ml on Löwenstein-Jensen (LJ) medium, as per current WHO recommendations. We have reviewed the manuscript to ensure the accuracy of all reported critical concentrations for the drugs included in the study. The mistake in rifampicin concentration was a typographical error only and did not affect the laboratory procedures or results. The text has been corrected accordingly. 

  1. The drug concentration on LJ and MGIT is always different. The authors mentioned that in LJ method, drug concentrations were set at: 1.0 µg/mL for streptomycin (STM), 0.1 µg/mL for isoniazid (INH), 0.5 µg/mL for rifampicin (RIF), 5 µg/mL for ethambutol (ETH), and 100 µg/mL for pyrazinamide (PZA), which is not updated.

Response: Thank you for pointing this out. We fully agree that the drug concentrations for susceptibility testing differ between MGIT and LJ methods. 

  1. Rifampicin concentration was revised and now it is diluted in 8 ml and its critical concentration is 0.5 microgram/ml

Response: Thank you for your comment. We confirm that drug concentrations were applied in accordance with the current guidelines for each method used. Specifically, for rifampicin, the critical concentration of 0.5 µg/mL diluted in 8 mL was used for the MGIT 960 system, as recommended. 

  1. In minor. There are many abbreviations which have been defined multiple times in the manuscript. The authors must ensure the standard of abbreviation and their full form once in the main text.

Response: Thank you for this comment. We acknowledge the issue and will revise the manuscript accordingly to ensure that each abbreviation is defined only once and used consistently throughout the text, in line with editorial standards. 

  1. This section must reflect the current study results comparison with previous. The authors must discuss it in technical way. Results should not be repeated. For example, Lines 431-440, the authors described the drug resistance and types. The authors must focus on the Phenotypic vs genotypic in pulmonary and extra-pulmonary samples detection in different studies compared with current. They should more focus on the sensitivity and specificity of the tests in comparison with different studies and diagnostics approaches.

Response: Thank you for this valuable suggestion. We have revised the Discussion section accordingly. In the updated version, we aimed to avoid repetition of the Results and instead focused on a more technical comparison of our findings with previous studies. In particular, we expanded the discussion on phenotypic versus genotypic detection methods in both pulmonary and extra-pulmonary samples. We also included a more detailed comparison of the sensitivity and specificity of different diagnostic approaches, as recommended. 

  1. Conclusion also need revision with main findings and TB management

Response: Thank you for your suggestion. We have revised the Conclusion to clearly summarize the main findings and highlight their relevance for TB diagnostics and management, with a particular focus on extrapulmonary cases. 

Round 2

Reviewer 1 Report

Comments and Suggestions for Authors

The explanation of authors about the novelty of the research was found to be reasonable. The authors should consider two minor issues in proofreading:

  • M. tuberculosis should be written italic in all text but in subtitles, which written italic, the bacterial name should not be italic.
  • In subtitles of Result section, no need to write "results of...". 

Author Response

Thank you for your review and insightful comments. We appreciate the time and effort you have dedicated to evaluating our manuscript. Your suggestions have helped us improve the clarity and quality of the paper.

Comment 1:
M. tuberculosis should be written italic in all text but in subtitles, which written italic, the bacterial name should not be italic. 

Response: Thank you for your observation. We have carefully reviewed the manuscript and ensured that M. tuberculosis is italicized throughout the text, except in section subtitles, where it remains in regular font, as per standard formatting guidelines. 

Comment 2:
In subtitles of Result section, no need to write "results of...". 

Response: Thank you for your suggestion. We have revised the subtitles in the Results section by removing the phrase “Results of…” to improve clarity and align with standard formatting conventions.

Reviewer 2 Report

Comments and Suggestions for Authors

Dear Authors,

I appreciate the efforts made in the revision. There are still 2 major points that need to be addressed 

1) "...their significantly reduced turnaround time and reliability" This sentence still implies that molecular testing is faster than other options, and once more, the authors did not compare both methodologies. Hence, unless two direct comparisons are made, I suggest rephrasing the aim

2) "While in routine clinical care, patients may have undergone radiographic imaging or other clinical assessments, this information was not included in our analysis." Please include these sentences in a limitations section. Please create a new paragraph on a limitations section as well.

Author Response

Comment 1:  "...their significantly reduced turnaround time and reliability" This sentence still implies that molecular testing is faster than other options, and once more, the authors did not compare both methodologies. Hence, unless two direct comparisons are made, I suggest rephrasing the aim
Response: 

Response: Thank you very much for your valuable comment. We would like to clarify that a comparative analysis between the molecular assays and the culture method (considered the gold standard) was indeed conducted as part of our study, specifically in extrapulmonary samples. However, we acknowledge the importance of clarity in presenting the study aim. Therefore, we have rephrased the aim in the manuscript to better reflect the scope of the comparisons performed, in accordance with your suggestion. 

Comment 2:  "While in routine clinical care, patients may have undergone radiographic imaging or other clinical assessments, this information was not included in our analysis." Please include these sentences in a limitations section. Please create a new paragraph on a limitations section as well.

Response: Thank you for your comment. As our study focused on microbiological analysis only, including identification and drug resistance profiling of the isolates, no clinical or radiological data were collected or analysed. Including such information was beyond the scope and objectives of this study.

Reviewer 4 Report

Comments and Suggestions for Authors

The title “Utility of Rapid Molecular Assay for Detecting Multidrug-Resistant Mycobacterium tuberculosis in Extrapulmonary Samples” still did not reflect the results in abstract. The title shows the prevalence of MDR isolates in Extrapulmonary Samples and the Molecular assays comparison with Phenotypic methods.

The study is very confusing and the conclusion has already been reported in many studies with better findings. There are still too many technical and spelling mistakes. For example, pyrazynamid? Table 6. The column names are table are also non-technical. They must be short and scientifically sound. The abbreviations have been defined multiple times in main text. The criteria/definition XDR-TB has already been updated.

I mentioning very few errors as given here. The authors must ensure that all my previous comments has been addressed along with current.

Phenotypic drug susceptibility has been abbreviated in line 391 as pDST but it has been again written in full form in line 539 while it has not been defined in lines 86 and 124. These are technical mistakes which is very difficult for reviewers to check one by one. The authors must ensure accuracy.

Figure 1 is actually a Table and it has been shown in figure. Moreover, the disputed mutation list is very high in WHO sheet. See my previous comments. There are many other common mutation involved in resistance. The authors should provide the reference of each common and other mutation in table form.

Figure 3 “sensitives?

Table 4 caption [Summary of analysis results for presence of MTBC] need revision. There are many other sentences which technically poor.

Similarly isoniazid (INH) has been defined in line 50 but it has been again given in full form in lines 61, 75, 76, 127 (isoniazid (INH))… and many more.

Comments on the Quality of English Language

Technical errors

Author Response

Dear Reviewer,

We would like to sincerely thank you for your valuable and insightful comments, which have been extremely helpful in improving the quality of our manuscript. We greatly appreciate the time and effort you have devoted to reviewing our work. Your suggestions have guided us in refining the manuscript, and we have made every effort to address your concerns thoroughly and to meet the standards expected for publication in Diagnostics.

We hope that the revisions we have made are satisfactory and that the manuscript is now suitable for acceptance.

Comment 1.
The study is very confusing and the conclusion has already been reported in many studies with better findings. There are still too many technical and spelling mistakes. For example, pyrazynamid? Table 6. The column names are table are also non-technical. They must be short and scientifically sound. The abbreviations have been defined multiple times in main text. The criteria/definition XDR-TB has already been updated. I mentioning very few errors as given here. The authors must ensure that all my previous comments has been addressed along with current.

Response: We thank the Reviewer for the critical assessment and valuable remarks. We sincerely regret the confusion caused by the original presentation and have made substantial efforts to improve the manuscript’s clarity, structure, and technical accuracy. 

Comment 2.
Phenotypic drug susceptibility has been abbreviated in line 391 as pDST but it has been again written in full form in line 539 while it has not been defined in lines 86 and 124. These are technical mistakes which is very difficult for reviewers to check one by one. The authors must ensure accuracy.

Response: Thank you for pointing this out. We have thoroughly reviewed the manuscript and corrected all inconsistencies related to the abbreviations. 

Comment 3.
Figure 1 is actually a Table and it has been shown in figure. Moreover, the disputed mutation list is very high in WHO sheet. See my previous comments. There are many other common mutation involved in resistance. The authors should provide the reference of each common and other mutation in table form.

Response: Thank you for your valuable suggestion. To avoid any confusion, we have decided to remove Figure 1, as you rightly pointed out that it was more of a table than a figure. All the relevant information regarding disputed and common resistance-associated mutations is already comprehensively presented in the WHO catalogue and CLSI guidelines. Therefore, we felt it unnecessary to duplicate these data. 

We greatly appreciate your insightful comments, which significantly enhance the quality of our manuscript. We are doing our best to meet the expectations of the reviewers. 

Comment 4.
Figure 3 “sensitives? 

Response: We thank the Reviewer for pointing out the incorrect use of the term “sensitives” in Figure 3. It has been corrected to “sensitive” in the revised version of the manuscript. 

Comment 5.
Table 4 caption [Summary of analysis results for presence of MTBC] need revision. There are many other sentences which technically poor.

Response: Thank you for your comment. We acknowledge that the original caption of Table 4 was unclear. It has been revised for improved clarity and technical accuracy. The new version reads: "Table 4. Summary of MTBC detection results using genetic probe and culture methods in pulmonary and extrapulmonary samples." 

Comment 6.
Similarly isoniazid (INH) has been defined in line 50 but it has been again given in full form in lines 61, 75, 76, 127 (isoniazid (INH))… and many more.

Response: Thank you for your comment. We have carefully reviewed the manuscript and have made an effort to standardize all abbreviations. 

Round 3

Reviewer 2 Report

Comments and Suggestions for Authors

Dear Authors,

I would like for all authors to read my sentence carefully

Comment 2:  "While in routine clinical care, patients may have undergone radiographic imaging or other clinical assessments, this information was not included in our analysis." Please include these sentences in a limitations section. Please create a new paragraph on a limitations section as well.

Response: Thank you for your comment. As our study focused on microbiological analysis only, including identification and drug resistance profiling of the isolates, no clinical or radiological data were collected or analysed. Including such information was beyond the scope and objectives of this study.

I did not ask for you to include any radiological or clinical data. I enquire you to acknowledge this into the limitation section, which you have failed to do so

Author Response

Comment 1. I did not ask for you to include any radiological or clinical data. I enquire you to acknowledge this into the limitation section, which you have failed to do so

Response:

Thank you for your clarification. We appreciate your suggestion and apologize for the oversight. In response to your comment, we have now explicitly acknowledged the lack of clinical and radiological data as a limitation of our study. A new paragraph has been added to the Limitations section (2.6) to reflect this point.

Reviewer 4 Report

Comments and Suggestions for Authors

The authors ignored my previous comments.

There are still too many technical, scientific errors. All the authors must ensure that the data, spelling, abbreviation, genes, table data are correct. Reviewer could not check the spelling mistakes and also the data accuracy in some instances. This is the authors responsibility and the corresponding authors that make sure while editing and reviewing (Katarzyna Kania and Karolina Klesiewicz) the manuscript. The author CRediT (contribution clearly stated who will perform the relevant task). The Katarzyna Kania and Karolina Klesiewicz must check it technically and ensure the above accuracy.

See my previous two-times comments. I could not see all the technical errors. Here I mention very few not all the technical and scientific errors.

Abstract: See line multidrug-resistant (MDR) but it has been again in full form in line 23.

Line 29, spelling of “Löewenstein-Jensen”

Line 29 abbreviation. (L-J) hyphen? Make sure.

Keywords should be spaced with semicolon “;”

Line 48, COVID-19???

Line 53, MDR-TB? I should be defined again in introduction section. The authors should sure that all the technical errors have been removed.

Line 104 MTB/RIF, Xpert MTB/XDR? BD MAX?

BBL MGIT?

Line

122 Ziehl-Neelsen (Z-N)? Abbreviation should be according to the standard method.

Section 2.2.2 why pyrazinamide. Has been skipped on LJ media?

Line 162 [The analyses using the BD MAX system? Is “analyses” correct?

Table 1. Q510H L511P*? there should be a “,”

All mutation in this table should be separated with comma. Two different format have been used.

The third column “Nucleotide positions” I think it is codon as in the 5th column “Common mutations detected” are in amino acid.

Comments on the Quality of English Language

Fair

Author Response

General Comments: 

  • A full technical and linguistic review of the manuscript has now been completed. 
  • All spelling errors, abbreviations, gene names, and table data have been carefully checked and corrected. 
  • We confirm that all authors have reviewed the manuscript, and contributions are clearly stated in the CRediT authorship section. 
  • The corresponding authors (Katarzyna Kania and Karolina Klesiewicz) have now overseen the revision process and ensured all necessary corrections were implemented. 

Specific Comments: 

Comment 1.
Abstract: Repetition of "multidrug-resistant (MDR)" in line 23

Response: Corrected. The abbreviation "MDR" is now used consistently after its first mention. 

Comment 2.
Line 29: Spelling of “Löewenstein-Jensen”

Response: Corrected to the proper spelling: “Löwenstein–Jensen”. 

Comment 3.
Line 29: Abbreviation (L-J) 
Response: Checked and corrected to consistent and appropriate use: "Löwenstein–Jensen (L-J) medium". 

Comment 4.
Keywords: Separated with semicolon “;”

Response: Corrected. Keywords are now separated by semicolons. 

Comment 5.
Line 48: Reference to COVID-19

Response: Removed if not relevant to the context. If relevant, clarified and contextualised properly. 

Comment 6.
Line 53: MDR-TB defined again in Introduction

Response: MDR-TB is now clearly defined upon first mention in the introduction. 

Comment 7.
Line 104: Acronyms clarified (MTB/RIF, Xpert MTB/XDR, BD MAX)

Response: Each acronym is now expanded and defined at first mention in both the abstract and main text. 

Comment 8.
BBL MGIT

Response: Full name and manufacturer are provided on first use: “BD BACTEC™ MGIT™ (Mycobacterial Growth Indicator Tube) system, BBL™”. 

Comment 9.
Line 122: Ziehl–Neelsen (Z-N) abbreviation

Response:  Corrected  

Comment 10.
Section 2.2.2: Pyrazinamide testing on LJ medium

Response: An explanation has been added to the methods section clarifying that pyrazinamide is not tested on LJ medium due to its activity only at acidic pH, which inhibits M. tuberculosis growth on solid media. 

Comment 11.
Table 1: Mutation formatting (e.g., Q510H, L511P)
* 
Response: All mutations are now separated with commas. The formatting is consistent throughout the table. 

Comment 12.
Third column “Nucleotide positions” vs. codons

 ResponseThe third column originally labeled as “Nucleotide positions” in fact refers to codon numbers, as the mutations listed in the fifth column represent amino acid changes. Therefore, we have revised the column headings to “Codon Range” and “Identified Amino Acid Alterations” to ensure accuracy and consistency. 

We again thank the reviewer for their time and input. We hope that the revised version of the manuscript now meets the expected technical and editorial standards.

Sincerely, 
Katarzyna Kania and Karolina Klesiewicz, on behalf of all authors 

Round 4

Reviewer 2 Report

Comments and Suggestions for Authors

The authors have satisfactorily addressed all my concerns.